# SecVulEval: Benchmarking LLMs for Real-World C/C++ Vulnerability Detection

## Abstract

Large Language Models (LLMs) have shown promise in various software engineering tasks, but evaluating their effectiveness in vulnerability detection remains challenging due to the lack of high-quality benchmark datasets. Most existing datasets are limited to function-level labels, ignoring finer-grained vulnerability patterns and crucial contextual information. They also often suffer from poor data quality, such as mislabeling, inconsistent annotations, and duplicates, which can lead to inflated performance and weak generalization. Moreover, by including only the vulnerable functions, these datasets miss broader program context, like data/control dependencies and interprocedural interactions, that are essential for accurately detecting and understanding real-world security flaws. Without this context, detection models are evaluated under unrealistic assumptions, limiting their practical impact.

To address these limitations, this paper introduces SecVulEval, a comprehensive benchmark designed to support fine-grained evaluation of LLMs and other detection methods with rich contextual information. SecVulEval focuses on real-world C/C++ vulnerabilities at the statement level. This granularity enables more precise evaluation of a model's ability to localize and understand vulnerabilities, beyond simple binary classification at the function level. By incorporating rich contextual information, SecVulEval sets a new standard for benchmarking vulnerability detection in realistic software development scenarios. This benchmark includes 25,440 function samples covering 5,867 unique CVEs in C/C++ projects from 1999 to 2024. We evaluated the SOTA LLMs with a multi-agent-based approach. The evaluation on our dataset shows that the models are still far from accurately predicting vulnerable statements in a given function. The best-performing **Claude-3.7-Sonnet** model achieves **23.83% F1-score** for detecting vulnerable statements with correct reasoning, with GPT-4.1 closely behind. We also evaluate the effect of using contextual information for the vulnerability detection task. Finally, we analyze the LLM outputs and provide insights into their behavior in vulnerability detection for C/C++.

## 1 Introduction

Security vulnerabilities are a major concern for the safety and robustness of software systems. Much of the technological infrastructure in today's world heavily relies on C/C++ projects, and consequently, these projects are critical targets for security vulnerabilities. Vulnerabilities in these projects can have a widespread impact on many downstream systems, making their reliability and robust maintenance of paramount importance Lin et al. (2023); Gu et al. (2023). However, existing tools and techniques for detecting security vulnerabilities in C/C++ often fail to address real-world complexity, diverse codebases, and evolving security threats Shiri Harzevili et al. (2024). The rapid adoption of Large Language Models (LLMs) in software engineering has opened new avenues for automating many critical tasks Zhou et al. (2024); Guo et al. (2024b). While LLMs have demonstrated impressive potential in code-related tasks, their effectiveness in tackling real-world C/C++ security vulnerabilities remains underexplored.

As more and more LLMs emerge, a reliable benchmark is crucial to evaluating LLMs' capability to detect security vulnerabilities in C/C++ projects. Recently, many benchmarks have been proposed

for C/C++, such as BigVul Fan et al. (2020), CVEFixes Bhandari et al. (2021), DiverseVul Chen et al. (2023), MegaVul Ni et al. (2024), PrimeVul Ding et al. (2024), etc. Although promising, the existing benchmarks suffer from a few major limitations. First, they lack essential features such as statement-level vulnerability localization, which poses a significant challenge for tasks that require fine-grained analysis, training, or evaluation. Second, some datasets omit crucial details, like bug-fix code pairs, vulnerability types (CWE), and precise CVE metadata. The absence of this information limits researchers' and developers' ability to conduct in-depth investigations or build effective repair tools, ultimately hindering advancements in the field. Third, existing datasets frequently include only the vulnerable functions, omitting the broader program context that is essential for accurately identifying and understanding security flaws. This missing context encompasses critical aspects such as data and control dependencies, interprocedural interactions, and environment constraints, all of which play a key role in determining whether a piece of code is truly vulnerable and how the vulnerability manifests. Finally, although some datasets offer line-level labels (e.g., BigVul Fan et al. (2020)), simply having added-deleted lines from a commit may not be very useful. Specially, for C/C++, some statements can be very long, and it is common practice to break them down into several lines[1]. Therefore, it is hard to make a meaningful understanding of only the line fragments without seeing the entire statements.

Most of the current vulnerability detection techniques (e.g., Fu & Tantithamthavorn (2022), Hin et al. (2022), Sheng et al. (2024)) conduct vulnerability detection at a local scope, often focusing on a given function in isolation. These approaches frequently overlook critical contextual information from related codebases, such as variable state returned from an external function, function arguments, execution environment, etc. A recent study Risse & Böhme (2024) has shown through empirical evaluation that most vulnerabilities in C/C++ require some level of external context to be correctly identified, such as variables, functions, type definitions, and environmental constraints that affect the function. As a result, neglecting the contextual information of a code snippet hinders these techniques from accurately assessing the presence of vulnerabilities within the code. Their study further reveals that many of the machine learning (ML) techniques that report high scores in vulnerability detection may be learning spurious features instead of the true vulnerability. This underscores the need for a more granular identification of vulnerabilities, along with correct reasoning to determine whether the models can truly spot vulnerabilities.

In this work, we address these limitations by introducing a comprehensive C/C++ vulnerability dataset that provides granular information up to the statement level. We focus on vulnerabilities that have been patched, ensuring that the dataset reflects real-world fixes. For each vulnerability, we gather detailed metadata, including CWE types, corresponding CVE (Common Vulnerabilities and Exposures) IDs and descriptions, commit IDs, commit descriptions, changed files, changed functions, and the modified statements that were deleted or added before and after the patch. We also extracted the contexts related to the vulnerable functions using GPT-4.1 and added them to the dataset.

We adopt the five levels of context defined by Risse et al. Risse & Böhme (2024) to represent the essential context for a vulnerability: **Function Arguments**, **External Functions** (functions called from within the target function), **Type Execution Declarations** (e.g., struct, enum, and other type definitions), **Globals** (such as global variables and macros), and **Execution Environments** (e.g., the presence of a specific file in a given path). Our manual analysis of a subset of 1k (around 10% of the whole dataset) samples shows that GPT-4.1 can identify the contexts necessary for a given vulnerability in a function with 83.16% accuracy. The statement-level granularity and contextual information, along with other metadata, enable deeper analysis and a more accurate evaluation of vulnerability detection.

We further evaluate five LLMs, including both open-source models such as *Qwen2.5-Coder-32B*, *Deepseek-Coder-33B*, *Codestral-22B* and proprietary models like *GPT-4.1* and *Claude-3.7-Sonnet* on our dataset to show their ability to detect C/C++ vulnerabilities at statement level. Note that, in our experiments, we employ a multi-agent pipeline in which each agent is powered by an LLM. This design is motivated by prior work showing that decomposing complex tasks into smaller, actionable components can enhance LLM performance Li et al. (2024); Sreedhar & Chilton (2025); Tran et al. (2025), thereby justifying our choice of a multi-agent architecture. Our initial experiments show that state-of-the-art LLMs are still far from being applicable as vulnerability detection tools for C/C++.

---

[1] https://github.com/bminor/glibc/commit/2864e76#diff-70c1d4052e9de7ad76e1437a17f00b70a9ea6732a3667041ccfb8dbb0caccae4

Table 1: Comparison of SECVULEVAL to widely used C/C++ vulnerability datasets from key aspects, i.e., the number of vulnerable functions, the availability of metadata, the duplication rate, the availability of context information, and the detection level. Duplicate rates marked with * are reported from Ding et al. (2024).

| | #Vul Funcs | Metadata | Duplicate (%) | Context Info | Statement-Level |
|---|---|---|---|---|---|
| **DiverseVul** Chen et al. (2023) | 18,945 | ✗ | 3.3* | ✗ | ✗ |
| **ReVeal** Chakraborty et al. (2021) | 1,658 | ✗ | 1.54 | ✗ | ✗ |
| **Devign** Zhou et al. (2019) | 12,457 | ✗ | 0.26 | ✗ | ✗ |
| **CVEFixes** Bhandari et al. (2021) | 5,365 | ✓ | 18.9* | ✗ | ✗ |
| **BigVul** Fan et al. (2020) | 3,754 | ✓ | 12.7* | ✗ | ✗ |
| **SVEN** He & Vechev (2023) | 417 | ✓ | 0.44 | ✗ | ✗ |
| **PrimeVul** Ding et al. (2024) | 6,968 | ✓ | 0.0 | ✗ | ✗ |
| **MegaVul** Ni et al. (2024) | 8,254 | ✓ | 0.0 | ✗ | ✗ |
| SECVULEVAL | 10,998 | ✓ | 0.0 | ✓ | ✓ |

The top-performing *Claude-3.7-Sonnet* model attains only a 23.83% F1-score, with *GPT-4.1* trailing closely.

## 2 RELATED WORK ON VULNERABILITY DATASETS FOR C/C++

Zhou et al. Zhou et al. (2019) provides the *Devign* dataset, collected to evaluate their Devign detection model. This dataset includes over 12,457 C/C++ vulnerabilities. However, it does not include other metadata such as CWE, CVE, etc. Also, they collect the vulnerable functions with a simple commit search with string matching, which resulted in the inclusion of many inaccurate functions, i.e., up to 20% according to a manual analysis done by Risse & Böhme (2024) on a random subset. The *ReVeal* dataset proposed by Chakraborty et al. (2021) includes 1,658 C/C++ vulnerable functions, but only from the Chromium and Debian projects. Chen et al. Chen et al. (2023) proposed *DiverseVul*, a C/C++ vulnerability detection benchmark with 18,945 vulnerabilities (from 797 projects and covering 150 CWEs). They showed that code-specialized models, e.g., *CodeT5* and *NatGen*, surpass graph-based methods but face persistent issues such as high false positives, poor generalization, and limited data scalability, highlighting the need for improved deep learning approaches. Ding et al. Ding et al. (2024) introduced *PrimeVul*, another C/C++ vulnerability benchmark with rigorous de-duplication, chronological splits, and VD-S metrics, exposing code models' near-random failure despite prior overestimation and underscoring the urgency for innovative detection paradigms. However, all these datasets only include vulnerability annotations at the function level, i.e., whether the function is vulnerable or not. They lack vulnerability information at a more granular level, like the statement level. Statement-level labels are necessary to understand how the vulnerability is caused, which can then be utilized for better training and evaluation of vulnerability detection models.

Fan et al. Fan et al. (2020) proposed *BigVul*, a C/C++ vulnerability dataset derived from open-source GitHub projects containing 3,754 vulnerabilities from 348 projects to support vulnerability detection. This dataset is closer to our work, as it includes line-level labels for vulnerable functions. Bhandari et al. Bhandari et al. (2021) collected a large collection of 5,365 C/C++ vulnerabilities from NVD. It includes vulnerabilities at five levels of abstraction, including line-level vulnerability labels and other metadata. However, both datasets heavily suffer from high duplication rates, which risks data leakage in the testing or evaluation of detection models. The *SVEN* dataset proposed by He et al. He & Vechev (2023) also includes line-level vulnerability information, and has accurate labeling as the entire dataset is manually annotated. But, due to this manual process, the dataset only includes 417 vulnerable C/C++ functions, limiting its use cases. Moreover, these three datasets include line-level labels, which may not be useful in many cases. C/C++ is a verbose language, and it is common for a statement to span multiple lines. Therefore, a single line might only be a fragment of a statement, and therefore, by itself, does not carry meaningful information.

In our work, we address these shortcomings and challenges by including vulnerable and non-vulnerable functions along with statement-level vulnerability labels, along with contextual information for the vulnerability. These are accompanied by other metadata for varied analysis, along

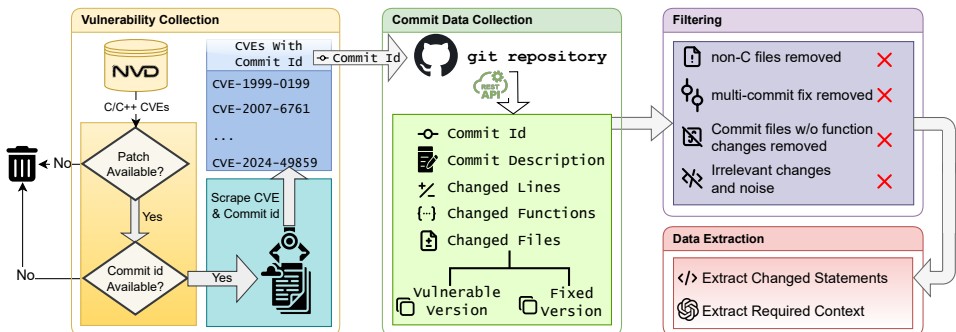

Figure 1: Data collection pipeline of SECVULEVAL

with rigorous de-duplication and filtering to maintain data quality. Table 1 provides a detailed comparison between our benchmark and previous works.

## 3 BENCHMARK CONSTRUCTION

In this section, we provide a detailed overview of the different steps and stages in building our benchmark data, i.e., vulnerability collection, commit data collection, noisy data filtering, and contextual information collection. The workflow is illustrated in Figure 1.

### 3.1 VULNERABILITY COLLECTION

We start by collecting CVEs recorded in the National Vulnerability Database (NVD)[2]. NVD has a rich collection of vulnerabilities and is regularly updated, making it a standard vulnerability repository. NVD provides detailed metadata for each CVE entry, including descriptions, severity scores (e.g., CVSS), affected products, and references to patches or advisories. However, one limitation of the NVD is that it does not explicitly categorize vulnerabilities by programming language. To focus specifically on C/C++ vulnerabilities, we leverage the project names identified as C/C++ projects in prior vulnerability datasets, such as BigVul Fan et al. (2020), CVEFixes Bhandari et al. (2021), and PrimeVul Ding et al. (2024), as listed in Table 1. By reusing these curated project names, we ensure that the collected CVEs are associated with C/C++ codebases. We further retrieve CVE records for these projects (called 'product' in NVD) where the CVE status is not *REJECTED*. We also utilize the keyword search feature of the NVD API to search with related keywords ('C++', 'C language', '.cpp', etc.). These results are then filtered by the file types, and only C/C++ vulnerabilities are kept. To enable the study of actual vulnerabilities, only CVEs with patch-related information are retained. Using the "Patch" tag from the NVD Developers API, CVEs without patch references are discarded. Additionally, to avoid duplication, we only kept the CVEs that had at least one link to a patch commit in their references. However, some of the commit links point to many forked repositories. We discarded such forked commits and only kept commits to the original repo. We ended up with a collection of CVEs, each with a description, associated CWE, fixing commit ID, and other metadata.

### 3.2 COMMIT DATA COLLECTION

Given the commit IDs collected for each CVE, the next step is to collect the commit-related information. We utilize the GitHub REST API to fetch commit details. For each commit, we collect its commit ID, commit message, and the files changed in the commit. We also sanitize the commit descriptions by removing accreditation lines (such as reporter emails, cc emails, etc.) since they do not contain any information related to vulnerable code changes. For each changed file, we extract the changed lines (i.e., added lines and deleted lines) and changed functions, i.e., functions containing the changed lines. For all changed files, lines, and functions, we include two copies: one *before* the fixing commit (i.e., vulnerable version) and one *after* the fixing commit (i.e., fixed version).

---

[2]https://nvd.nist.gov

Table 2: Top 10 projects in SECVULEVAL.

| Projects | CWEs | CVEs | Vul. Funcs | Non-vul. Funcs |
|---|---|---|---|---|
| Linux | 69 | 1,920 | 2,793 | 3,288 |
| Chrome | 35 | 526 | 1,748 | 1,940 |
| TensorFlow | 34 | 355 | 547 | 959 |
| Android | 23 | 209 | 551 | 749 |
| ImageMagick | 27 | 172 | 208 | 354 |
| vim | 21 | 162 | 182 | 290 |
| gpac | 22 | 138 | 151 | 215 |
| tcpdump | 8 | 109 | 145 | 206 |
| radare2 | 16 | 103 | 168 | 214 |
| FFmpeg | 20 | 99 | 110 | 152 |
| Total (707 Projects) | 145 | 5,867 | 10,998 | 14,442 |

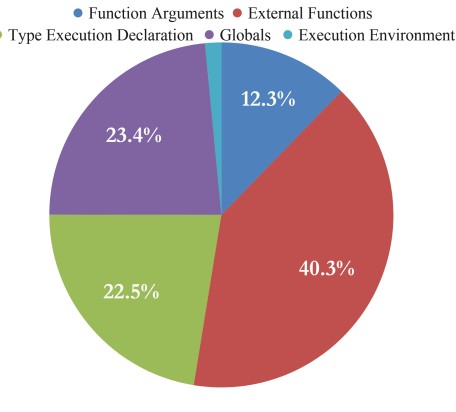

Figure 2: Distribution of five context categories in SECVULEVAL.

## 3.3 FILTERING & DE-NOISING

After collecting the commit artifacts, we apply a series of filtering and denoising steps to enhance the quality and reliability of our dataset. The filtering criteria are ① removing non-C files, ② removing multi-commit fixes, ③ removing commits with no functions changed, and ④ using heuristics to remove refactoring, reformatting, etc. Below, we describe them in detail.

① We exclude non-C/C++ files (e.g., *.S*, *.rst*, config files) as their changes are often in documentation, macros, or signatures, and are side effects unrelated to the vulnerability. ② We retain only CVEs with single-commit fixes for simplicity, as multi-commit data can be challenging to present effectively to the model. Our manual investigation revealed that, in many cases, multi-commit fixes primarily involve similar changes across multiple files or refactoring of related functions in other files. To maintain clarity and consistency, we discard only 43 CVEs with multiple commits. ③ We exclude commit files that do not involve any changes to functions. Some commits may only modify function prototypes, add comments, or update enum values or struct fields[3]. While these changes may be related to the codebase, they provide minimal insight into the actual vulnerability and instead introduce unnecessary noise. ④ Finally, we use heuristics to filter out tangled files in changes and improve the labeling accuracy. Specifically, when a commit changes several functions in a file, all the changes are not necessarily related to the vulnerable code. In fact, in many cases, variable/function renaming or function signature updating is carried out across the whole file, resulting in multiple functions being updated. We filter out tangled changes by retaining only functions (i) solely modified in the file, or (ii) explicitly referenced in the CVE or commit message, avoiding unrelated edits like renaming or formatting.

The final step in the filtering process eliminates any duplicate functions. Duplicates can arise for several reasons in the initial collection process, such as multiple copies of the same function in different files, or a CVE being assigned to multiple CWE types, etc. Duplicate entries are a big problem in vulnerability benchmarks as they can leak data to training and also make the benchmark biased towards excessive duplicates. Previous benchmarks such as DiverseVul Chen et al. (2023), BigVul Fan et al. (2020), and CVEFixes Bhandari et al. (2021) suffer from 3.3%-18.9% function duplication problem as shown in Ding et al. (2024). Our filtering process eliminates duplicate functions by mapping each function to an *md5* hash as done by Ding et al. (2024). We normalize each function by stripping away all leading/trailing whitespaces, '\n', and '\t'. Then we convert the function string to an *md5* hash and keep only one copy of a function in the case of a collision. In this way, we ensure that all functions in our dataset are unique, eliminating the data leakage problem.

After filtering, we end up with a collection of 5,867 unique C/C++ vulnerabilities (CVEs) from 707 different projects, distributed over 145 CWE types. Figure 3 shows the number of vulnerable functions from the top-20 CWE types. The benchmark consists of 25,440 functions, including 10,998 vulnerable and 14,442 non-vulnerable functions. The functions range in various sizes from 4 lines to 541 lines (from 2.5 to 97.5 percentile), with a median size of 44 lines per function. The

---

[3]https://github.com/torvalds/linux/commit/4e8771a3, https://github.com/torvalds/linux/commit/c06cfb0

average number of statements changed in each function is around 4 (deleted) and 6 (added). Table 2 shows the overall statistics of the dataset along with the top 10 most vulnerable projects.

### 3.4 CONTEXTUAL INFORMATION COLLECTION

Real-world vulnerabilities are intricate and often result from the interaction of multiple entities. However, previous works do not incorporate this important feature into their datasets. Indeed, it is extremely arduous to manually check each function in the dataset and identify the required contexts. To this end, we harness the code-analyzing ability of LLMs to automatically extract required contexts for vulnerable functions.

We prompt GPT-4.1 with all the available information to identify the context required to understand the vulnerability in this function, and categorize them according to the five definitions. We provide the following information to the model: *the vulnerability type* (CWE-ID and de-

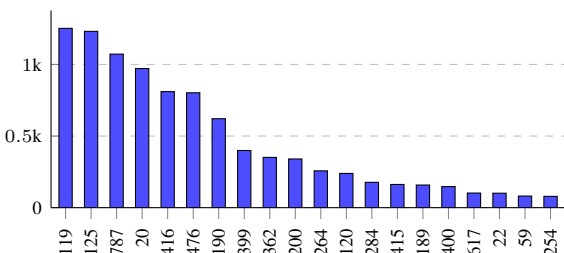

Figure 3: Number of vulnerable functions in Top-20 CWE types in SECVULEVAL. The x-axis represents the list of CWE IDs, while the y-axis indicates the number of corresponding samples.

scription), *the full function body*, *the patch* (deleted-added lines), *the commit message*, and *the CVE description*. Using this detailed information, the model decides which symbols (variables, functions, etc.) are required or help to identify the vulnerability in the given function.

To measure how accurately GPT-4.1 can identify related contexts in a given function, the authors worked together and manually validated 1k randomly selected samples (around 10% of the whole dataset). Ground truth contexts were determined by tracing variables used in vulnerable statements back to their external symbols or verifying their origin within the function, with external symbols being defined as the relevant contextual elements. Heuristics were also used to capture additional potentially influential symbols (e.g., those that affect branching within `if` or `switch-case` statements). A prediction was deemed correct if GPT-4.1 identified the ground truth contexts, permitting the inclusion of up to one superfluous symbol within any single category. Samples where no relevant context could be identified from the function (e.g., only hard-coded strings changed in the function) were marked as 'N/A' and excluded from the evaluation, resulting in ten such cases out of the initial 1k samples. Among the remaining samples with identifiable context, GPT-4.1 achieved an accuracy of $83.16 \pm 6.93\%$ (with 99% Confidence Interval). Incorrect predictions resulted primarily from missing one or two required symbols or incorrectly including unrelated ones, particularly when analyzing larger code modifications. Figure 2 shows the distribution of the five context categories in the dataset.

## 4 EXPERIMENTS

Comprehensive and diverse datasets are vital for vulnerability research. SECVULEVAL offers detailed, statement-level vulnerability annotations, enriched with contextual data and metadata like CWE IDs and CVE descriptions, enabling fine-grained, context-aware analysis. With 707 projects and 145 CWE types, it provides a diverse, fully de-duplicated benchmark ideal for evaluating detection techniques. This section uses SECVULEVAL to evaluate LLMs' effectiveness in detecting vulnerabilities (Section 4.1) and identifying essential contextual information (Section 4.2).

### 4.1 EXPERIMENT 1: VULNERABILITY DETECTION

We investigate the effectiveness of LLMs in detecting security vulnerabilities in C/C++ code. Previous studies have shown that single LLMs perform very poorly on the C/C++ vulnerability detection task, even when doing a function-level binary classification (vulnerable or non-vulnerable) Ding et al. (2024). Therefore, we adopt a multi-agent pipeline for vulnerability detection as illustrated in Figure 4. These LLM-based agents have separate responsibilities and complete the entire task through collaboration. This type of approach has been shown to be more effective than a single

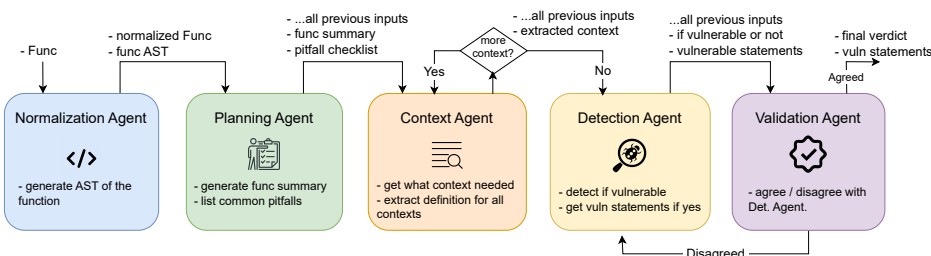

Figure 4: The multi-agent vulnerability detection pipeline.

LLM by multiple studies Li et al. (2024); Tran et al. (2025); Sreedhar & Chilton (2025). To the best of our knowledge, this is the first time that an LLM-based multi-agent pipeline is applied for the vulnerability detection task.

The pipeline consists of five agents, four powered by LLMs. It starts with the *Normalization Agent*, which parses the input function into AST form using tree-sitter. This output, along with the normalized function, is passed to the *Planning Agent*, where an LLM summarizes the function and generates a checklist of potential vulnerabilities. The *Context Agent* then iteratively queries an LLM to identify required external symbols for vulnerability detection, stopping once context is deemed sufficient or after three attempts. Symbol definitions are extracted via tree-sitter also. The *Detection Agent* uses all prior inputs to determine if the function is vulnerable, pinpoint vulnerable statements, and provide a rationale. Finally, the *Validation Agent* evaluates the *Detection Agent*'s output. If disagreement arises, the *Detection Agent* reruns (up to three iterations) until both agents agree.

## 4.2 EXPERIMENT 2: CONTEXT IDENTIFICATION

The second experiment demonstrates the use of our dataset for context identification, i.e., evaluating how effectively LLMs extract the contextual elements required for vulnerability detection. Within the *Context Agent*, an LLM is prompted to identify relevant symbols, such as function arguments, external calls, and type definitions, that are needed to analyze a target function and determine the presence of vulnerabilities. These identified definitions are then forwarded to the *Detection Agent*. To assess the accuracy of context extraction, we compare the LLM-generated symbols against the ground-truth dependencies annotated in our dataset.

## 4.3 EVALUATION METRICS

In this work, we use *Precision*, *Recall*, and *F1-Score* to measure the vulnerability detection performance of the models. Specifically, vulnerable instances are regarded as positive, and non-vulnerable instances as negative. For statement-level vulnerability detection, we measure a prediction as True Positive if it correctly predicts vulnerable statements along with accurate reasoning. If the model correctly identifies a function as vulnerable, but the reasoning is incorrect, we consider this a True Positive for function-level detection but a False Negative for statement-level detection since it misses the correct vulnerability. Precision measures the likelihood that a prediction is correct when it identifies an instance as positive. Recall, on the other hand, measures the ability to correctly identify all the positive instances, even if the model makes some False Positives. Finally, F1-Score is the harmonic mean of both Precision and Recall.

## 4.4 MODEL SETUP

We selected five state-of-the-art LLMs for our evaluation tasks, including open-source and proprietary models. Specifically, we select *Deepseek-Coder-33B-Instruct* Guo et al. (2024a), *Codestral-22B-v0.1* Jiang et al. (2023), *Qwen2.5-Coder-32B-Instruct* Hui et al. (2024), *GPT-4.1*, and *Claude-3.7-Sonnet* because these models are widely used in the community and have demonstrated high performance in various software engineering tasks. For the open-source models, we have used the weights from HuggingFace.

Table 3: Vulnerability detection performance of LLM-driven agents on Func-level (whether the function is vulnerable or not) and Stat-level (if the model identified vulnerable statements in the function with correct reasoning)

| Models used in the agents | Precision (%) | | Recall (%) | | F1-Score (%) | |
|---|---|---|---|---|---|---|
| | Func-Level | Stat-Level | Func-Level | Stat-Level | Func-Level | Stat-Level |
| Qwen2.5-Coder-32B | 35.48 | 12.10 | 34.92 | 15.46 | 35.20 | 13.57 |
| Deepseek-Coder-33B | 45.16 | 5.65 | 46.28 | 9.72 | 45.71 | 7.14 |
| Codestral-22B | **47.41** | 12.93 | 45.83 | 18.75 | 46.61 | 15.31 |
| GPT-4.1 | 40.65 | 14.49 | 73.11 | 49.21 | 52.25 | 22.38 |
| Claude-3.7-Sonnet | 41.86 | **15.35** | **75.63** | **53.23** | **53.89** | **23.83** |

We report all LLM scores with pass@1 and use $temperature = 0.1$ for stable outputs as commonly used in the literature DeLorenzo et al. (2024); Jain et al. (2023). We use pass@1 as it is more representative of the real scenario where a developer does not have the output reference to validate the attempts. All our experiments were carried out in a system with Intel(R) Xeon(R) Gold 6442Y CPU and a GPU cluster of 4 NVIDIA L40S.

## 5 RESULT ANALYSIS

### 5.1 EXPERIMENT 1: BENCHMARKING LLMS IN VULNERABILITY DETECTION

**Approach:** To find the effectiveness of LLMs in detecting statement-level vulnerabilities, we adopted a multi-agent based approach as described in Section 4.1. We run our evaluation on the output of the Validation Agent. If the model finds any vulnerability, then it shall output each vulnerable statement and its reason as a pair. Otherwise, the model should output an empty list, and 'is_vulnerable' field in the output will be 'false'. Since the outputs include explanations for the vulnerability, it is not possible to automatically evaluate the results. In addition, the statements returned by the model may not always be the exact same statements as the changed statements. For example, sometimes the models output the 'sink' statement as vulnerable (where the vulnerability causes a crash or other symptoms) instead of the 'source' (where the vulnerability is introduced), or sometimes return both statements. Moreover, sometimes the model may return the correct statements, but with incorrect reasoning. Therefore, we randomly selected 1k samples (around 10% of the whole dataset) for the experiment and manually validated the outputs. We identify an output to be True Positive for statement-level detection if: ① the model outputs the exact vulnerable statements and the correct reasoning, ② if same vulnerability is fixed at multiple places in the function, the model at least returns one such statement with correct reasoning, ③ the model returns the vulnerable statements with correct reasoning and at most two unrelated statements, ④ the model outputs only the sink statements or both vulnerable and sink statements with correct reasoning. For function-level detection, we automatically match the output of the 'is_vulnerable' field with the ground truth from the dataset.

**Results:** Table 3 shows that statement-level detection performance remains low. The best model, Claude-3.7-Sonnet, achieves only a 23.83% F1-score and 15.35% precision, with GPT-4.1 close behind. Open-source models perform worse overall. Closed-source models like Claude and GPT-4.1 adopt a more aggressive detection strategy, reflected in their higher recall (e.g., Claude's 53.23% vs. Codestral's 18.75%) but lower precision due to more false positives. This trend is even more pronounced at the function level, where Claude and GPT-4.1 show very high recall (75.63%, 73.11%) but still lower precision than open-source models, suggesting a tendency to over-flag functions as vulnerable.

Note that when comparing the performance of these models on function-level and statement-level vulnerability detection in Table 3, we can observe that identifying whether a function is vulnerable and pinpointing the exact vulnerable statements are distinct tasks, the latter being significantly more challenging with lower precision and recall values. Both open-source and proprietary LLMs show significant drops in precision and recall when required to locate vulnerable statements with correct root cause explanations. Unlike function-level detection, statement-level analysis demands fine-grained reasoning about issues like pointer arithmetic and memory bounds, requiring deeper

contextual and interprocedural understanding. This raises concerns about the reliability of function-level predictions. *Risse et al.* Risse & Böhme (2024) have shown that models may rely on spurious patterns rather than true vulnerabilities. Our results echo this, as models often fail to find the real vulnerability even when correctly flagging a function. Thus, advancing statement-level detection is essential, as without it, developers risk being misled by false positives or missing subtle but critical flaws. We show the breakdown performance per CWE in Table 5 in the Appendix C. We also conduct an ablation study to examine LLM performance in detecting vulnerabilities without agent support, with results provided in Appendix D. Overall, most models achieve F1-scores below 4%, underscoring the motivation for our agent-based approach to accurate vulnerability detection.

> The models are not effective at detecting vulnerable statements in C/C++ functions, as the best performing *Claude-3.7-Sonnet* model achieves only 23.83% F1-score. SECVULEVAL's diverse set of vulnerabilities uncovers LLMs' severe lack of ability to find vulnerable statements and their root cause in complex real-world code.

## 5.2 EXPERIMENT 2: BENCHMARKING LLMS IN CONTEXT IDENTIFICATION

**Approach:** This experiment evaluates how well LLMs identify the contextual elements needed for vulnerability detection. Within the *Context Agent*, an LLM is prompted to extract relevant context, such as function arguments, external calls, and type definitions, for analyzing a target function. To evaluate the performance of LLMs in this task, we focus on vulnerabilities for which the *Context*

Table 4: Accuracy of essential context identification. No models identified any Environment-level context.

| Models | Func Args | Ext. Func | Type Def. | Globals |
|---|---|---|---|---|
| Qwen2.5-Coder-32B | 2.80% | 27.25% | 20.00% | 24.23% |
| Deepseek-Coder-33B | 5.31% | 29.80% | 13.97% | 19.81% |
| Codestral-22B | 0.00% | 27.52% | 6.96% | 6.60% |
| GPT-4.1 | 3.51% | 27.52% | 20.00% | 18.87% |
| Claude-3.7-Sonnet | 0.0% | 37.00% | 41.33% | 34.53% |

*Agent* determined that additional contextual information was needed to facilitate detection. For these cases, we compare the context extracted by the LLMs against the ground-truth dependencies provided in our dataset.

**Results:** Results show that LLMs struggle to identify key contextual information needed to understand function-level vulnerabilities. Most models, except Claude-3.7-Sonnet, primarily focus on external functions, which are useful but insufficient. Globals and type definitions, which include macros, constants, and structs, are also critical in C/C++ but often overlooked, limiting the models' contextual understanding. Claude-3.7-Sonnet performs better in identifying these elements, likely contributing to its higher detection score. All models perform poorly on identifying function arguments, possibly due to focusing on their struct types rather than the variables themselves. None of the models identified any environment-level context. This is understandable as it is very rare (1.5% of all contexts), and models are unlikely to catch environmental contexts from the function.

> The results reveal the limited capability of current LLMs in identifying relevant contextual information for vulnerability detection. While Claude-3.7-Sonnet demonstrates relatively higher coverage, other models frequently overlook critical context types.

## 6 CONCLUSION

This paper presents SECVULEVAL, a novel dataset for C/C++ vulnerability detection that emphasizes statement-level granularity and rich contextual information. We collected 5,867 unique CVEs from C/C++ projects spanning 1999 to 2024 in NVD. Leveraging SECVULEVAL, we evaluated five state-of-the-art LLMs on two tasks: vulnerability detection and context identification. The results show that current LLMs perform poorly in detecting vulnerabilities, with Claude-3.7-Sonnet achieving the highest F1-score of 23.83%. Claude-3.7-Sonnet also outperformed others in identifying the contextual information necessary for detection. With its fine-grained labels and contextual annotations, SECVULEVAL provides a valuable benchmark for training and evaluating models on real-world vulnerability detection at the statement level.

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

## A    APPENDIX

The appendix includes the following five sections:

- LLM usage

- Performance breakdown based on CWE types

- Performance of LLMs in vulnerability detection without agent support

- Limitation

- Broader Societal Impact

- Failure Analysis of LLMs on Vulnerability Detection

- Prompts

- Resources & Execution Time

## B    LLM USAGE

In this paper, we leverage LLMs to assist in proofreading and improving writing quality. The models help identify and correct issues such as grammar errors, unclear phrasing, awkward sentence structures.

## C    PERFORMANCE BREAKDOWN BASED ON CWE TYPES

Table 5: Precision (P), Recall (R), and F1 scores of the models for statement-level vulnerability detection in each CWE type. It shows the top-10 most frequently detected CWE types. The CWE types where the models could not correctly detect any statement are marked with '-'.

| Models | | CWE-119 | CWE-125 | CWE-787 | CWE-189 | CWE-190 | CWE-476 | CWE-362 | CWE-369 | CWE-401 | CWE-416 |
|---|---|---|---|---|---|---|---|---|---|---|---|
| Qwen2.5-Coder-32B | P | **20.0** | 13.6 | 10.0 | - | 14.3 | **15.4** | - | - | 50.0 | 20.0 |
| | R | 30.0 | 30.0 | 12.5 | - | 8.3 | 28.6 | - | - | 50.0 | 12.5 |
| | F1 | 24.0 | 18.8 | 11.1 | - | 10.5 | 20.0 | - | - | 50.0 | 15.4 |
| Deepseek-Coder-33B | P | - | 5.9 | 10.0 | 33.3 | 14.3 | 8.7 | - | - | - | - |
| | R | - | 9.1 | 16.7 | 50.0 | 50.0 | 22.2 | - | - | - | - |
| | F1 | - | 7.1 | 12.5 | 40.0 | 22.2 | 12.5 | - | - | - | - |
| Codestral-22B | P | - | 7.1 | **20.0** | 66.7 | 28.6 | 12.5 | - | - | **66.7** | 5.9 |
| | R | - | 10.0 | 28.6 | 66.7 | 66.7 | 33.3 | - | - | 100.0 | 9.1 |
| | F1 | - | 8.3 | **23.5** | **66.7** | 40.0 | 18.2 | - | - | **80.0** | 7.1 |
| GPT-4.1 | P | 17.4 | 8.8 | 13.6 | 40.0 | 27.3 | 13.6 | - | 33.3 | 50.0 | 23.1 |
| | R | 57.1 | 75.0 | 75.0 | 100.0 | 75.0 | 100.0 | - | 100.0 | 100.0 | 17.6 |
| | F1 | **26.7** | 15.8 | 23.1 | 57.1 | 40.0 | **24.0** | - | 50.0 | 66.7 | 20.0 |
| Claude-3.7-Sonnet | P | 13.0 | **13.8** | 14.3 | 40.0 | **33.3** | 9.8 | **20.0** | 66.7 | 66.7 | **31.3** |
| | R | 42.9 | 50.0 | 60.0 | **100.0** | 100.0 | 100.0 | 66.7 | 100.0 | 100.0 | 33.3 |
| | F1 | 20.0 | **21.6** | 23.1 | 57.1 | **50.0** | 17.8 | **30.8** | **80.0** | **80.0** | **32.3** |

Table 5 reports model performance per CWE type. We present the ten most frequently occurring CWEs, as the remaining categories appear too infrequently to support statistically meaningful comparisons. As shown, several CWEs (CWE-125, CWE-787, CWE-189, CWE-190, CWE-476, CWE-401, and CWE-416) are detected by most models, whereas CWE-362 and CWE-369 are identified only by Claude-3.7-Sonnet and GPT-4.1. Although CWE-401 occurs relatively rarely in the dataset, it is detected reliably by most models. In contrast, CWE-20 is the fourth most common vulnerability (see Figure 3) yet is detected only by GPT-4.1. Likewise, CWE-120 and CWE-415 appear with moderate frequency but are not detected by any model, suggesting that these vulnerabilities are more challenging for current systems to recognize.

Table 6: Statement-Level Vulnerable Detection Performance of LLMs

| Model Name | Precision(%) | Recall(%) | F1-Score(%) |
|---|---|---|---|
| Deepseek-Coder-33B | 0.28 | 0.24 | 0.26 |
| Codestral-22B | 1.12 | 0.67 | 0.84 |
| Qwen2.5-Coder-32B | **2.24** | **22.33** | **4.07** |
| GPT-4 | 1.98 | 5.23 | 2.87 |

# D  PERFORMANCE OF LLMS IN VULNERABILITY DETECTION WITHOUT AGENT SUPPORT

For the ablation study, we also evaluated LLM performance on statement-level vulnerability detection without agent support. For cost considerations, we limited this experiment to GPT-4.1 (noting that GPT-4.1 is significantly cheaper than Claude-3.7-Sonnet in terms of token usage). The results are presented in Table 6. As shown, model performance falls far short of practical usability. The best-performing model in terms of F1-score, *Qwen2.5-Coder-32B*, achieved only 4.08, with a precision of 2.25%. *GPT-4* followed with an F1-score of 2.88%. Other open-source models, including *CodeLlama*, *DeepSeek-Coder*, and *Codestral*, all recorded F1-scores below 1%. The relatively high recall of *Qwen2.5-Coder-32B* suggests that the model tends to over-predict vulnerabilities, often labeling benign lines as vulnerable. This results in a large number of false positives, which limits its practical applicability and further motivates our agent-based solution for vulnerability detection.

# E  LIMITATION

**Context Collection:** The contextual information integrated into our dataset was automatically extracted by an LLM. While a manual validation performed on 1k random samples indicated an accuracy of $83.16 \pm 6.93\%$ for correctly identified contexts, this assessment method has inherent limitations. The very definition of "relevant context" is not an objective quantity, leading to potential inter-annotator variability and dependence on subjective human interpretation. Additionally, the dataset does not extend to repository-wide contexts, such as transitive effects of non-local interactions beyond immediate function calls and domain-specific knowledge of the projects.

**Manual Analysis:** Despite leveraging available information, e.g., patch, commit description, CVE description, and issue tracking details (if available), manual validation poses several methodological limitations. The inherent subjectivity in defining precise vulnerability characteristics can lead to variability in judgments. To mitigate this, we define specific criteria (as detailed in our approach in Section 5.1 in the main paper). In addition, manual analysis is a labor-intensive process that imposes significant scalability constraints.

**Model variability:** We evaluated the largest size of each model that we could run with our resources. We could not investigate intra-model size variability since running different sizes of the same model significantly increases the experiment scale and is not feasible with our resources and time constraints. Therefore, the same conclusion may not hold for the same models of smaller sizes and needs further exploration.

# F  BROADER SOCIETAL IMPACT

**Positive:** The dataset includes a friendly license, making it widely available for both research and commercial use. Moreover, insights into how context improves detection will steer future tool design toward deeper program understanding. Stronger tools built on these findings can surface flaws earlier in critical C/C++ code, reducing exploit risk. This increased efficiency can boost productivity and scale the overall software development process.

**Negative:** However, as with any vulnerability dataset, this dataset can be used by malicious users to train models to try to exploit vulnerabilities.

Table 7: Size comparison between the detected and missed functions. The reported *p-values* are obtained from one-sided Mann-Whitney U tests; significance is evaluated at the 5% level.

|  | Claude-3.7-Sonnet | GPT-4.1 | Qwen2.5-Coder-32B | DeepSeek-Coder-33B | Codestral-22B |
|---|---|---|---|---|---|
| Detected Vulnerable | 33 | 31 | 18 | 14 | 25 |
| Missed Vulnerable | 71 | 71 | 65 | 64 | 67 |
| p-values | 0.0168 | 0.0115 | 0.000582 | 0.000197 | 0.000538 |

# G   FAILURE ANALYSIS OF LLMS ON VULNERABILITY DETECTION

## G.1   EFFECTS OF FUNCTION SIZE

To determine whether the function length has distinctive effects on the performance of the LLMs, we performed a thorough analysis. We partitioned a complete set of 120 true-positive functions from a 300-sample set into those correctly flagged by each model and those it failed to detect. Then, we compared the two groups' lines of code (LoC) distributions. For each model, we applied a one-sided Mann-Whitney U test to each model independently, assessing the hypothesis that detected functions are much shorter than missed ones. Across all models, the medians for detected functions ranged from 14 LoC to 33 LoC, whereas the medians for missed functions lay between 64 LoC and 71 LoC; the corresponding U-tests yielded p-values from $5.38 \times 10^{-4}$ to $1.68 \times 10^{-2}$, as shown in Table 7. These results consistently reject the null hypothesis of identical length distributions at the 5% significance level, showing that every model is significantly more successful at recognizing vulnerabilities in compact, shorter functions and consistently misses them in the longer ones.

## G.2   INCLINATION TO SPECIFIC TYPES

Our manual validation revealed a tendency for the models to exhibit excessive false positives for certain vulnerability types. Specifically, Null Dereference (CWE-476), Use-After-Free (CWE-416), and Integer Overflow (CWE-190) were frequently over-detected. GPT-4.1 and Claude-3.7-Sonnet, in particular, consistently misidentified numerous pointer dereferences as Null Dereference vulnerabilities, even when explicit null checks were present. For example, as illustrated in Figure 5, despite `pep->ring` being checked in line-6 to handle null cases, GPT-4.1 erroneously flagged its dereference in line-10 as a Null Dereference vulnerability.

```c
int cdnsp_endpoint_init(struct cdnsp_device *pdev,
    struct cdnsp_ep *pep,
    gfp_t mem_flags)
{
    // large function. previous lines skipped
    if (!pep->ring)
        return -ENOMEM;
    // other statements. pep->ring is not affected here.
    ep_ctx->deq = cpu_to_le64(pep->ring->first_seg->dma |
        pep->ring->cycle_state);
    // large function. next lines skipped
}
```

Figure 5: GPT-4.1 marking line-10 as Null Dereference but `pep->ring` is checked in line-6.

Furthermore, the models often erroneously classified memory deallocation calls, e.g., `free()`, `kfree()`, as Use-After-Free vulnerabilities without adequately validating subsequent pointer usage. In one instance, GPT-4.1's reasoning, "The function `git_free` frees the `tmp_path` object, making any subsequent access to `tmp_path` a potential use-after-free vulnerability." was flawed, as the function returned immediately, precluding any later use of `tmp_path`. Finally, many arithmetic operations were flagged as Integer Overflow (CWE-190) without sufficient analysis to determine if an overflow or wraparound condition could occur.

These tendencies for over-detection were particularly pronounced in GPT-4.1 and Claude-3.7-Sonnet, which exhibited a more aggressive vulnerability flagging behavior. From the set of 300 samples, comprising 120 genuinely vulnerable functions, GPT-4.1 and Claude-3.7-Sonnet identified

214 and 215 functions as vulnerable, respectively. In contrast, Qwen2.5-Coder-33B, Deepseek-Coder-33B, and Codestral-22B demonstrated a more conservative approach, flagging only 121, 124, and 116 functions as vulnerable, which resulted in a significantly lower incidence of false positives.

## H  PROMPTS

Here we describe the prompts used for our four LLM-based agents. During initial experimentation, we noticed the models intermittently refusing to process the prompts, interpreting them as attempts to exploit security vulnerabilities. We redesigned the prompts, explicitly emphasizing *defensive* security research and objectives against exploitation, which successfully mitigated the issue.

PLANNING AGENT

**System:**

You are a senior C/C++ *defensive* security researcher.
Your task: devise an initial **analysis plan** to detect whether a vulnerability exists in the function supplied by the user.
Respond with **JSON only** - no markdown, no explanations outside the JSON. Required keys:
 "summary": $\leq$ 3-sentence plain-language overview of the function.
 "checklist": array ($\leq$ 8) of concise red-flag cues (code patterns, data-flow hints, etc.) that indicate the existence of a vulnerability.
Constraints:
- Focus exclusively on *detection* and *prevention* - do NOT produce exploitation steps.
- Do not invent symbols that are absent from the code or AST.
- Prefer brevity; omit speculative items if unsure.

**User:**

=== Function Source Code (could be truncated if too long) ===
{function code}
=== Compact AST JSON (could be truncated if too long) ===
{ast}

CONTEXT AGENT

---

**System:**

You are assisting in *defensive* C/C++ vulnerability analysis.
Given the target function and the context already collected, decide whether additional external symbols are needed to detect a vulnerability. External symbols can be functions, macros, global variables, typedefs, structs, or enums.
Return JSON ONLY, schema:
{
   "need_more": bool,
   "new_symbols": [ {name, kind}, ... ]
}
If need_more is false, "new_symbols" must be an empty array. List at most 6 symbols and never repeat ones already provided. You cannot get indefinite context from the user, so choose the most important context you need for vulnerability detection in this function.

---

**User:**

=== Function Source Code (could be truncated if too long) ===
{function code}
=== Existing Context Symbols ===
{context}
=== Checklist Hints ===
{checklist}

---

DETECTION AGENT

**System:**

You are a C/C++ security analyst.
Decide whether the given function contains a vulnerability *defensively* (no exploit instructions). If vulnerable, list the minimal set of source statements that directly cause the issue. Only flag as vulnerable if you are absolutely sure. Not everything needs checks, so be sure to only identify the ones that can cause issues according to the other contexts given to you. Return JSON ONLY, schema:
{
  "is_vulnerable": bool,
  "vuln_statements": [
    {"line": <0-based>, "statement": "<raw text>", "reason": "<why>"}
  ]
}
Return an empty array if "is_vulnerable" is false.

**User:**

### Function (line-numbered): ###
{line numbered function}
### AST (could be truncated if very long): ###
{ast}
### Function summary from Planning Agent: ###
{summary}
### Common Pitfall checklist for this function: ###
{checklist}
### Helpful external context definitions: ###
{context}

VALIDATION AGENT

**System:**

You are a senior C/C++ security reviewer.
Given the code, AST, planning cues, and the previous Detection Agent's verdict whether vulnerable or not, decide whether you **agree**.
If you disagree, provide your own corrected list of vulnerable statements. Only flag as vulnerable if you are absolutely sure. Not everything needs checks, so be sure to only identify the ones that can cause issues according to the other contexts given to you.
Return JSON ONLY, schema:
{
  "agree": bool,
  "is_vulnerable": bool,
  "vuln_statements": [
      {"line": <0-based>, "statement": "<raw text>", "reason": "<why>"}
  ]
}
Keep "vuln_statements" empty if "is_vulnerable" is false.

**User:**

### Function (line-numbered): ###
{line numbered function}
### AST (could be truncated if very long): ###
{ast}
### Function summary from Planning Agent: ###
{summary}
### Common Pitfall checklist for this function: ###
{checklist}
### Helpful external context definitions: ###
{context}
### Detection Agent Output: ###
{detection output}

# I    RESOURCES & EXECUTION TIME

All models were run on a multi-GPU cluster of 4 NVIDIA L40S with Intel(R) Xeon(R) Gold 6442Y CPU and 250 GiB RAM. Table 8 presents the average execution time of each agent. The model weights are used from HuggingFace and implemented using the `transformers.pipeline` architectures. The time for Context Agent includes the model inference time and the symbol extraction (fetching definition of functions, typedef, etc., with tree-sitter) time. The time for Context Agent and the Detection+Validation Agents can also be affected by the number of retries, i.e., more retries will increase the average time per function. We only show the execution time for the open-source models since the execution time for GPT-4.1 and Claude-3.7-Sonnet models is subject to different rate limits affecting total execution times and will not provide a fair comparison.

Table 8: Average execution time (time per function) for each agent for the open-source models. The Normalization is model agnostic and was executed only once for all models. The Detection and Validation were run together as they interact between themselves and hence are shown together here.

| Model | Normalization Agent (ms) | Planning Agent (s) | Context Agent (s) | Detection+Validation Agent (s) | Total (min) |
|---|---|---|---|---|---|
| Qwen2.5-Coder-32B | | 52.33 | 96.19 | 213.26 | 6.03 |
| Deepseek-Coder-33B | 1.2 | 59.49 | 137.16 | 283.01 | 7.99 |
| Codestral-22B | | 36.25 | 59.54 | 202.17 | 4.97 |

