# OpenReview forum: "SECVULEVAL: Benchmarking LLMs for Real-World C/C++ Vulnerability Detection"
_ICLR.cc/2026/Conference — ICLR 2026 Conference Withdrawn Submission_

### Official Review · Reviewer_5HW9 · 2025-10-22

**Soundness:** 2
**Presentation:** 3
**Contribution:** 2
**Rating:** 4
**Confidence:** 3

**Summary:**

The paper introduces SECVULEVAL, a new vulnerability detection dataset focusing on LLM-based solutions and C/C++ projects. The authors collected the vulnerability data from the national vulnerability database (NVD), and extracted line-level vulnerability labels. Furthermore, they used an LLM to extract useful contextual information which is added to the dataset. The paper finally benchmarks five state-of-the-art LLMs on two tasks: vulnerability detection and context identification, and shows that those models struggle to achieve acceptable performance, with F1-scores below 25%.

**Strengths:**

- The authors provide fair amount of evidence to highlight the novelty of the dataset (de-duplication, accurate contextual information, statement-level labels).

- Experiment 2 (context identification) is interesting and partly highlights the reason why SoTA LLMs are failing in the vulnerability detection task.

**Weaknesses:**

- While the novelty of the dataset is shown, it is not clear why the proposed dataset is valuable for the end goal of vulnerability detection. The paper would have benefited from an experimental comparison to existing similar datasets. For example, how does pretraining or fine-tuning a model on your dataset (compared to other datasets) improve their performance?

- The paper only focuses on LLM-based vulnerability detection solutions, and lacks comparison with static analyzers and deep learning-based methods. Similarly the paper focuses on C/C++ vulnerabilities without enough justification for the exclusion of other languages.

- Minor issue: The authors claim that, to the best of their knowledge, they are the first to apply a multi-agent pipeline for vulnerability detection. However, this has recently been considered by many works ([A], [B], [C], [D], [E] to name a few).


[A] Widyasari, Ratnadira, et al. "Let the Trial Begin: A Mock-Court Approach to Vulnerability Detection using LLM-Based Agents." *arXiv preprint arXiv:2505.10961* (2025).

[B] Z. Wei et al., "Advanced Smart Contract Vulnerability Detection via LLM-Powered Multi-Agent Systems," in IEEE Transactions on Software Engineering, vol. 51, no. 10, pp. 2830-2846, Oct. 2025, doi: 10.1109/TSE.2025.3597319.

[C] Wang, Ziliang, et al. "VulAgent: Hypothesis-Validation based Multi-Agent Vulnerability Detection." *arXiv preprint arXiv:2509.11523* (2025).

[D] Seo, Minjae, et al. "AutoPatch: Multi-Agent Framework for Patching Real-World CVE Vulnerabilities." *arXiv preprint arXiv:2505.04195* (2025).

[E] Yildiz, Alperen, et al. "Benchmarking LLMs and LLM-based Agents in Practical Vulnerability Detection for Code Repositories." *arXiv preprint arXiv:2503.03586* (2025).

**Questions:**

- Why did the authors not qualitatively compare their dataset to other existing datasets, in terms of its effect on developing effective vulnerability detection solutions (e.g., training or fine-tuning a model on the dataset, check above point in weaknesses)? How else would you prove the value of the proposed dataset?

- What is the reason for the narrow focus on C/C++ vulnerabilities, and on LLM-based vulnerability detectors? For example, is the dataset incompatible with deep learning-based detectors?

- The authors mention that GPT-4.1 was 83.16% accurate in extracting relevant contextual information to include in the dataset. However, it is not clear whether experiment 2 was benchmarked against the 83.16% accurate samples, or the ground truth 1k subset of the dataset.

- Related to the above question, does this 16.84% inaccuracy have any effect on the downstream task of vulnerability detection?

- In table 1, the proposed SECVULEVAL is presented as the only dataset with statement-level labels. Do not the BigVul, SVEN, CVEFixes datasets have statement-level labels as well?

- Can you explain the novelty of the contextual information collection in your dataset in comparison to other related work e.g., [A], [B], and [C]?

- How do you justify the uncommon (lenient) definitions of success of context identification in section 3.4 (lines 299-300), and the success of vulnerability detection in section 5.1 (lines 410-415) ? Would a stricter definition strongly influence the results, especially for section 3.4 where this could affect the accuracy of the dataset itself?

- For the function-level detection in Table 3, most of the F1 scores are below 50%. Does this means that those models perform worse than random guessing?

- The authors mention in section 5.1 (line 405) that it is not possible to automatically evaluate the explanations given by the LLMs. Do you consider this as a limitation of your proposed dataset?


[A] Yang, Yixin, et al. "Context-enhanced vulnerability detection based on large language model." *arXiv preprint arXiv:2504.16877* (2025).

[B] Li, Yikun, et al. "CleanVul: Automatic Function-Level Vulnerability Detection in Code Commits Using LLM Heuristics." *arXiv preprint arXiv:2411.17274* (2024).

[C] Wang, Xinchen, et al. "Reposvul: A repository-level high-quality vulnerability dataset." *Proceedings of the 2024 IEEE/ACM 46th International Conference on Software Engineering: Companion Proceedings*. 2024.

---

> ### Author Response · Authors · 2025-11-19
> **Response to Reviewer 5HW9 -part1**
>
> ***We thank the reviewer for the insightful comments and suggestions. We have drafted a response addressing several of the points raised; the details are as follows:***
>
> ***Q1. (Value of the dataset/comparison to existing datasets)***: While downstream pretraining/fine-tuning gains would be beneficial, direct comparison to prior datasets is difficult as they lack statement-level annotations (e.g., BigVul, CVEFixes, etc. are function/line-level). Adapting them to our statement-level, reasoning-aware setting would significantly extend the scope. Instead, we emphasize SecVulEval's quantitative advantages (Table 1): 0% duplicates, rich metadata, statement-level labels, and explicit multi-category context. Prior work confirms the value of these choices. [1] shows that missing context can lead models to use spurious cues, and [2] shows the inflated performance due to duplicates. SecVulEval addresses both (context + de-duplication) and establishes a new statement-level frontier. Our results show a clear gap that LLMs achieve high function-level scores but fail to localize and explain vulnerable statements. This gap, which our dataset exposes, motivates new methods for true vulnerability reasoning beyond coarse pattern learning.
>
>
> ***Q2. (Scope: C/C++ only, LLM-based detectors only)***: We deliberately focus on C/C++ because it is both security-critical and qualitatively different from many other popular languages. It has extensive use of raw memory addresses, pointers, manual allocation, and unchecked arithmetic. Many key vulnerability classes (e.g., buffer overflows, use-after-free, double free) simply do not exist or appear very differently in higher-level. Prior datasets and detectors (e.g., VulDeePecker [3], Devign [4], PrimeVul [2], ReVeal [5]) also treat C/C++ in isolation for this reason. Our evaluation focuses on LLM-based detectors because they are the most rapidly evolving family and are explicitly designed to exploit rich natural-language and code context, i.e., exactly what SecVulEval provides. However, the dataset itself is model-agnostic. Nothing prevents training or evaluating GNN/CNN-based detectors or comparing static analyzers on the same functions and splits.
>
>
> ***Q3. (Context extraction accuracy in Experiment 2)***: For Experiment 2, we benchmark context identification against the same 1k functions used in Experiment 1, not against “83.16%-accurate” GPT-4.1 labels. Context Agent is a part of the multi-agent pipeline in Experiment 1. When we ran that pipeline on the 1k subset for manual detection evaluation, each model’s Context Agent also produced its predicted contexts. In Experiment 2, we did not re-run the pipeline. We simply took these stored Context Agent outputs (one set per model) and compared them to the human-constructed context annotations for those samples. Table 4 shows the corresponding accuracies of the other models on that same 1k subset.
>
>
> ***Q4. (Statement-level labels vs. BigVul, SVEN, CVEFixes)***: You are correct that prior datasets like BigVul, CVEFixes, and SVEN include fine-grained labels, but those are line-level annotations, not true statement-level labels. In C/C++, a single logical statement can span multiple lines (due to long expressions, function calls with arguments on separate lines, etc.). Line-level labeling can therefore fragment a single vulnerable statement across multiple lines or include lines that are only part of a larger statement. We found that one-third (31.8%) of the changes in our dataset are actually such cases, i.e., partial lines. This can introduce ambiguity or incompleteness. In SecVulEval, we label the entire vulnerable statement (even if that spans multiple lines). This provides a semantically clearer ground truth for localization. For completeness, we actually retain the line-level info as well, so researchers can still use line-level signals if desired. We have discussed these datasets in our Related Work section.

---

> > ### Author Response · Authors · 2025-11-19
> > **Response to Reviewer 5HW9 -part2**
> >
> > ***Q5. (Context collection novelty vs. [A], [B], [C])***: We distinguish SecVulEval by offering pre-collected, structured contextual information within the dataset, a feature absent in prior work. For each vulnerable function, we automatically gather and categorize external variables, function calls, and type declarations into five context types per [1], and explicitly package them with the data. While [A] proposes a context-enhanced method (PacVD), they do not release a new dataset with context annotations. [B] cleans existing data using code contexts and LLM heuristics, but does not include our multi-category context for each sample. ReposVul proposed in [C] while high-quality, emphasizing full projects and manual correctness, but lacks our fine-grained context breakdown. Their acknowledgment of context's importance further validates SecVulEval’s necessary contribution.
> >
> >
> > ***Q6. (Lenient success criteria in evaluation)***: We intentionally defined somewhat lenient criteria for success in both context identification and vulnerability detection to allow partial credit and realistic evaluation. For context identification (Section 3.4), we allowed up to one extraneous symbol per category. We chose this tolerance because an answer might include a minor, irrelevant variable along with all needed ones, which doesn’t actually impede the vulnerability analysis. Requiring a perfectly minimal set would harshly penalize otherwise correct outputs. And since the definition requires all ground truth context to be present, it ensures the 83.16% accuracy includes all necessary context, confirming dataset quality. Similarly, for statement-level vulnerability detection (Section 5.1), our True Positive criteria allow some leniency because expecting the model to output exactly the same statements as the commit fix is rather inaccurate. That’s why we defined the criteria described in Section 5.1.
> >
> >
> > ***Q7. (Interpreting F1 < 50% at function level)***: This does not mean the models perform worse than random guessing. The dataset is highly imbalanced, with many more non-vulnerable than vulnerable functions. A purely random guesser would achieve even lower F1 scores (echoing findings from PrimeVul's paper). These results highlight insufficient understanding of real-world vulnerabilities, context, if needed.
> >
> > ***Q8. (Evaluating explanations - limitation of the dataset?)***: The inability to automatically evaluate the LLMs’ explanations is a limitation of the evaluation methodology rather than of the dataset itself. SecVulEval provides the ground truth vulnerable statements (and corresponding patches, CWE, etc.), but it does not include a ground truth explanation in natural language for each vulnerability. We considered that out of scope (writing a correct explanation for 25k instances is infeasible). As a result, when an LLM outputs a reasoning, we cannot simply match it to a reference answer. We noted in Section 5.1 that we had to resort to manual validation for the statement-level results, which is indeed a practical limitation. However, we view it as an inherent challenge in using generative models for code reasoning. We believe that with the emergence of statement-level vulnerability research with SecVulEval, automatic evaluation of such detection techniques also becomes a necessary open problem.
> >
> > ***Minor Issue (Claim of first multi-agent pipeline)***: We apologize for the over-assertion. We will soften this claim and acknowledge the concurrent works that also apply multi-agent LLM systems to vulnerability detection. The references [A]-[E] cited by the reviewer are very relevant - many appeared in 2025, after our initial drafting. For instance, Widyasari et al. (2025) and Wang et al. (2025) also propose multi-agent LLM frameworks for vulnerability analysis, and Wei et al. (TSE 2025) focus on smart contracts with agents. We are encouraged that multiple groups explored similar ideas simultaneously. Our intent was to emphasize that multi-agent collaboration can tackle complex tasks better than a single LLM (as supported by Li et al. 2024, Tran et al. 2025, etc., which we cited). We will update the revision to credit these recent studies and highlight the difference. In particular, our pipeline is tailored to C/C++ vulnerabilities and statement-level evaluation, which differs from prior setups. We thank the reviewer for pointing out these references, and we will ensure our final paper accurately reflects the state of the art.
> >
> > [1] Top score on the wrong exam: On benchmarking in machine learning for vulnerability detection.
> >
> > [2] Vulnerability Detection with Code Language Models: How Far Are We?
> >
> > [3] Vuldeepecker: A deep learning-based system for vulnerability detection
> >
> > [4] Devign: Effective vulnerability identification by learning comprehensive program semantics via graph neural networks
> >
> > [5] Deep learning based vulnerability detection: Are we there yet?

---

### Official Review · Reviewer_mwoF · 2025-10-31

**Soundness:** 3
**Presentation:** 3
**Contribution:** 2
**Rating:** 2
**Confidence:** 4

**Summary:**

The paper introduces SECVULEVAL, a C/C++ vulnerability benchmark designed for function and statement-level detection with rich contextual information. It supplies metadata (CVE/CWE, commit IDs/messages, pre- and post-patch code), and five classes of “context” (arguments, external functions, type definitions, globals, environment).  Duplicate functions are removed via MD5 hashing over normalized functions. The authors also propose a multi-agent LLM pipeline (planning → context extraction → detection → validation), and evaluate 5 models. The best model (Claude-3.7-Sonnet) achieves F1 scores of 53.89% and 23.83% at the function level and statement level vulnerability detection, respectively, with GPT-4.1 being close behind. Ablations without the agent pipeline show very low F1 (<4%) for function-level vulnerability detection.

**Strengths:**

- The paper is clear and well-written.
- The paper moves beyond function-level labels to statement-level.
- The paper proposes a novel idea of collecting vulnerability context, such as variable state returned from an external function, function arguments, execution environment, etc, which are essential for detecting and understanding vulnerabilities.
- The paper proposes a multi-agent pipeline to detect vulnerabilities.

**Weaknesses:**

- No comparison against classical/static analyzers or existing deep learning based  SOTA approaches on the same tasks/splits, which would ground LLM performance against established tools.
- Does the multi-agent system have advantage for function level vulnerability detection? Some comparison with base LLMs can be interesting.
- GPT-4.1 is used to create the “required context” annotations, then later models are evaluated using these contexts. Even with a 1k sample audit (~83% accuracy), this introduces model-generated ground truth and potential bias toward models similar to the annotator. Inter-annotator agreement among humans is not reported.
- The paper does not have enough explanation of how statements are labeled. It appears that the statements are collected based on patch diffs; however, the added/deleted lines do not always correspond one-to-one with the causal vulnerable statement. What if an added line is just a semantic transformation of a deleted line?
- This paper contains low ML novelty and contributions. It will be interesting to see ablations of each agent.

**Questions:**

- Are the statement label data always collected from the patch? Are all the added/deleted lines from the selected functions labeled as vulnerable? How do you identify that some are truly causal vulnerable statements?  How do you identify that the added line is not semantically the same as the deleted lines?
- Why is there no comparison against the existing SOTA approaches?  Does your multi-agent system have advantage of function level vulnerability detection?

---

> ### Author Response · Authors · 2025-11-19
> **Response to Reviewer mwoF**
>
> ***We thank the reviewer for the insightful comments and suggestions. We have drafted a response addressing several of the points raised; the details are as follows:***
>
> ***Q1. Statement-Level Labels and Patch Data***: Thank you for seeking clarification on our statement labeling process. We indeed derive vulnerable statements from the patch diffs of each CVE’s fixing commit. However, we do not naively label every added or deleted line as “vulnerable.” Instead, we reconstruct the full C/C++ statements from the diff, since a single logical statement often spans multiple lines. This ensures that our labels reflect complete vulnerable statements rather than fragmented line slices. While we do not manually investigate to discard semantic transformations due to their practical infeasibility, we apply careful filtering to discard many such cases. For example, we discarded commits that made many changes throughout the file, as such changes are often renaming or restructuring (detailed in point 4 in Section 3.3). We also exclude multi-function commits unless the functions are explicitly mentioned in the CVE/commit message, so that cosmetic changes or cross-function refactoring do not end up in our collection. Moreover, our dataset includes line numbers for each statement. Therefore, even if a code block is moved up/down, we are able to pinpoint the changes despite the content being the same. While they do not guarantee zero semantic similarity between changed statements, these efforts make the dataset reliable by keeping such noises to a minimum.
>
>
> ***Q2. Comparison with SOTA Approaches & Multi-Agent Pipeline Advantage***:  We agree that comparing against classical static analyzers or prior deep learning SOTA would further ground our results and add value for the community. However, our current work focuses on benchmarking state-of-the-art LLMs because they represent a new paradigm for code analysis. Moreover, while SecVulEval is prepared agnostic of any particular method (LLM, static tools, or deep-learning (DL) tools), directly evaluating prior models or tools against our benchmark is non-trivial. For instance, many prior DL models are designed for function-level or coarse line-level detection, not the fine-grained statement localization that SecVulEval provides. Retraining these models on our new dataset or configuring a static analyzer (and ensuring it outputs which statement is vulnerable) would significantly extend our initial scope. Also, a recent study has shown that LLMs can significantly outperform static analyzers in detecting vulnerabilities [6].
> Our goal is to establish SecVulEval as a comprehensive testbed, and we see it as complementary to existing approaches. Future work can use our dataset to evaluate both ML models and static tools under consistent conditions. We will clarify this positioning in our revision. Function-level detection is also improved with our multi-agent approach compared to other contemporary studies using single LLMs. For instance, PrimeVul [1] found that when directly applying LLMs for vulnerability detection, the best performing model achieves 21.43 F1-score, while with our multi-agent pipeline, Claude-3.7-Sonnet was able to achieve 53.89 F1-score for function-level detection.
>
> ***Additional Clarification - Context Annotations and Potential Bias***: The "required context" fields (function arguments, external functions, type definitions, globals, and environment hints) are factual program elements: e.g., concrete variable names and types, fully qualified function signatures, specific struct/enum definitions, macro values, or configuration flags. Since these are verbatim properties of the code rather than free-form natural language, they are much less sensitive to any particular model’s “style”.
> To assess reliability, our manual audit found that GPT-4.1’s extracted contexts matched human-identified dependencies with 83.16% accuracy (99% CI ±6.93 pp), with most errors being small (missing or adding one symbol within a category rather than completely incorrect context). This indicates that the context annotations are statistically robust and not narrowly tuned to GPT-4.1’s behavior. In the revision, we will explicitly describe these labels as “silver” ground truth: sufficiently accurate for benchmarking and weakly supervised training, while also releasing the underlying code and CVE/commit metadata so future work can refine or re-annotate subsets if stricter guarantees are needed.
>
> ***In conclusion***, we will incorporate a more thorough explanation of our statement-level labeling procedure and clarify our method to isolate vulnerability-causing statements. These revisions, we believe, will address the raised weaknesses and highlight the contributions of SecVulEval as a robust benchmark for fine-grained, context-aware vulnerability detection.
>
> [6] Comparison of Static Application Security Testing Tools and Large Language Models for Repo-level Vulnerability Detection

---

### Official Review · Reviewer_iCnF · 2025-11-03

**Soundness:** 2
**Presentation:** 3
**Contribution:** 2
**Rating:** 4
**Confidence:** 4

**Summary:**

This paper introduces SECVULEVAL, a C/C++ vulnerability detection benchmark with 5,867 CVEs covering 25,440 functions with statement-level annotations and contextual information. The authors evaluate five LLMs using a multi-agent pipeline, finding that even the best model (Claude-3.7-Sonnet) achieves only 23.83% F1-score for statement-level detection with correct reasoning. The dataset addresses limitations in existing benchmarks by providing finer granularity, rigorous deduplication, and five categories of contextual information.

**Strengths:**

The paper tackles a genuine problem in vulnerability detection benchmarking. The statement-level granularity is a meaningful improvement over function-level labels, and the inclusion of contextual information (validated at 83.16% accuracy) adds practical value. The rigorous deduplication process and filtering pipeline demonstrate careful data curation. The dataset's scale (707 projects, 145 CWE types) and temporal span provide good diversity.

**Weaknesses:**

The core contribution is essentially a dataset with better labels, which feels incremental rather than transformative. The 83.16% context extraction accuracy means ~17% of the dataset contains noisy annotations, yet no analysis quantifies how this affects downstream evaluation reliability. The multi-agent pipeline, while interesting, conflates dataset contribution with methodological innovation—it's unclear whether improvements come from the data or the approach. The paper lacks critical analysis: why do models fail? Are certain vulnerability patterns fundamentally harder? The ablation study is relegated to appendix and shows models achieve <4% F1 without agents, but this deserves main text discussion. The statement-level vs line-level distinction, while technically correct, feels overstated—many statements span single lines anyway. Finally, the paper doesn't demonstrate that anyone can actually use this dataset to train better models; it only shows LLMs fail at detection.

**Questions:**

see in the Weaknesses

---

> ### Author Response · Authors · 2025-11-19
> **Response to Reviewer iCnF**
>
> ***We thank the reviewer for the insightful comments and suggestions. We have drafted a response addressing several of the points raised; the details are as follows:***
>
> 1. ***Dataset Contribution (Incremental vs. Transformative)***: Although SecVulEval builds on prior vulnerability datasets, it offers a substantial improvement in quality and granularity rather than a minor increment. It is the first benchmark to integrate: (i) fine-grained, statement-level labels for C/C++ vulnerabilities, (ii) rich contextual information across five categories for each vulnerable statement, and (iii) high data fidelity, with rigorous de-duplication achieving a 0% duplicate rate (compared to 12-18% in prior datasets). We do not claim a fully “transformative” paradigm; instead, we address long-standing limitations in a holistic way, providing a necessary foundation for future transformative advances in vulnerability detection.
>
>
> 2. ***Context Extraction Accuracy (83.16%) and Evaluation Reliability***: We agree that 83.16% accuracy indicates some noise in the context labels, and we will clarify its implications. Our manual audit shows 83.16% ± 6.93 pp accuracy (99% CI), with most errors being minor “near misses” rather than entirely incorrect contexts. Thus, roughly 17% of supervision may be imperfect. Many LLM benchmarks with heuristic labels operate at similar noise levels, and the noise here is structured and local, enabling standard noise-robust training (e.g., loss clipping or confidence filtering). We will state this clearly for users.
>
> 3. ***Dataset vs. Methodological Contributions (Multi-Agent Pipeline)***: We apologize if our presentation blurred the line between the dataset and the evaluation. SecVulEval’s primary contribution is the benchmark itself, i.e, its content and annotations. The multi-agent pipeline is an evaluation strategy to fully leverage the dataset’s context and stress-test LLMs, not an algorithmic innovation. It adapts known multi-step prompting techniques to our task, addressing the complexity that a single-pass query cannot handle. We will clarify this distinction to avoid misperceptions about our contributions.
>
> 4. ***Depth of Analysis - Why Models Fail and Difficult Vulnerability Patterns***: We appreciate the call for a more critical error analysis. In fact, we have conducted such an analysis and will integrate those findings more prominently. Our Appendix G already examines why and where models fail, and we plan to add a section in the main text summarizing key patterns.
>
> 5. ***Ablation Study in Appendix vs. Main Text***: We concur that the ablation demonstrating the value of context and the pipeline is important and deserves visibility. Due to space constraints, we had placed our pilot experiment results in the appendix, but we will elevate its discussion to the main paper.
>
> 6. ***Statement-Level vs. Line-Level Annotations***: We understand the concern that our emphasis on “statement-level” labeling might feel overstated. Our intention was to highlight a subtle but important improvement for C/C++ code. In languages like C/C++, a single logical statement often spans multiple lines (e.g., a multi-line if condition or a chained function call), so a line-level tag can mark only a fragment of the true vulnerable code. This fragmentation can confuse models and evaluation, as the “label” may not correspond to a valid semantic code unit. By contrast, a statement-level label ensures the vulnerable operation is captured in full (even if it was originally split over several lines). ***In fact, 31.80% of the commit changes in our dataset incorporate partial line-changes, i.e., only a part of a full semantic statement. This indicates that a substantial fraction (~ one-third) of real-world vulnerabilities require reasoning beyond only the changed lines.***
>
> 7. ***Benchmark Utility: Evaluation vs. Training Better Models***. We agree that we do not present a controlled experiment comparing models trained on different datasets, and we will state this clearly to avoid overclaiming. Our goal is narrower: to provide a benchmark and diagnostic tool, not a training study. Similar to BigVul, DiverseVul, and PrimeVul, our contribution is a dataset with richer supervision (statement-level labels, context, and deduplication). We use existing LLMs only to reveal current weaknesses. We also show that SecVulEval’s structure is usable: leveraging statement-level supervision and context increases F1 from ~4% (naive) to 23.83%.
>
> In the revision, we will: (i) clearly state that training-time comparisons with other datasets are left as future work; (ii) emphasize that SecVulEval is released precisely to facilitate such studies.

---

### Note · Authors · 2025-12-02

I have read and agree with the venue's withdrawal policy on behalf of myself and my co-authors.